# The Effect of Netting Bags on the Postharvest Quality, Bioactive and Nutritional Compounds, and the Spoilage Microorganisms Content of Bell Peppers

**DOI:** 10.3390/foods12102071

**Published:** 2023-05-21

**Authors:** Selene C. H. Rives-Castillo, Zormy N. Correa-Pacheco, María L. Corona-Rangel, Mónica Hernández-López, Laura L. Barrera-Necha, Rosa I. Ventura-Aguilar, Silvia Bautista-Baños

**Affiliations:** 1Instituto Politécnico Nacional, Centro de Desarrollo de Productos Bióticos, Carretera Yautepec-Jojutla, km. 6, CEPROBI No. 8, San Isidro, Yautepec, Morelos 62731, Mexico; riveselene@gmail.com (S.C.H.R.-C.); zormynacary@yahoo.com (Z.N.C.-P.); mcorona@ipn.mx (M.L.C.-R.); mohernandezl@ipn.mx (M.H.-L.); lbarrera@ipn.mx (L.L.B.-N.); 2CONACYT-Centro de Desarrollo de Productos Bióticos, Instituto Politécnico Nacional, Carretera Yautepec-Jojutla, km. 6, CEPROBI No. 8, San Isidro, Yautepec, Morelos 62731, Mexico

**Keywords:** ‘California Wonder’, *Capsicum annuum* L., biodegradable, cactus stem residue, *Opuntia ficus-indica* Mill., packaging, PBTA, PLA, storage

## Abstract

The packaging nets used for bell peppers act as a form of protection. However, the manufacturing is based on polymers that cause serious environmental problems. The effects of nets made of biodegradable materials, such as poly(lactic) acid (PLA), poly(butylene adipate-co-terephthalate) (PBAT), and cactus stem residues, were evaluated on four different colors of ‘California Wonder’ bell peppers stored over a 25-day storage period under controlled and ambient temperature conditions. Compared to commercial polyethylene nets, the bell peppers kept in the biodegradable nets did not show notable differences with respect to color, weight loss, total soluble solids, and titratable acidity. However, there were significant differences (*p* < 0.05) in terms of phenol content, carotenoids (orange bell peppers), anthocyanins, and vitamin C, with an overall tendency to show a higher content in those kept in PLA 60%/PBTA 40%/cactus stem flour 3% compared to commercial packaging. In addition, the same net notably reduced the development of bacteria, fungi, and yeasts during the storage of red, orange, and yellow bell peppers. As postharvest packaging for bell peppers, this net could be considered a viable option for the storage of this product.

## 1. Introduction

Mexico is a worldwide exporter of bell peppers with an annual production of approximately 1.6 million tons. Bell peppers are an important source of health-promoting compounds. They are rich in bioactive compounds, including, iron, carotenoids, phenols, antioxidants, and vitamin C [1,2,3], and can be found in different colors, including red, yellow, orange, and green (https://www.healthline.com/nutrition/foods/bell-peppers, accessed on 1 April 2023).

They also require careful handling and storage due to their high susceptibility to loss of quality in terms of aspects such as water, color, and firmness [4]. In addition, fresh bell peppers cannot be stored for long periods due to their inherent compositional and textural characteristics. Consequently, relatively low-temperature storage conditions of about 8 to 10 °C remain the most effective tools for maintaining their quality and extending their shelf life. In addition, the use of packagings, such as boxes and nets, is very important for maintaining their freshness during storage [5].

The nets used in agricultural products during the postharvest stage generally serve for transport and act as protective packaging. Generally, the packaging materials are manufactured from petroleum-based polymers, such as polyethylene (PE) and polypropylene (PP), and hence are not biodegradable. This means that they eventually have a high environmental impact due to their accumulation in the environment [6,7]. 

For this reason, recently, the food packaging sector’s interest has been centered on developing eco-friendly and biodegradable packaging using biomaterials derived from renewable natural resources [8]. Among the sustainable proposals for the elaboration of packaging are biodegradable polymers, including polylactic (acid) (PLA) and poly (butylene adipate-co-terephthalate) (PBAT). These are some of the most commercialized polymers due to their high availability and low cost [9,10]. Along these lines, Musioł et al. [11] evaluated PLA and PBAT as a form of biodegradable packaging for possible use with fruits. Black-Solis et al. [12] also formulated a polymeric matrix of PLA/PBAT and cinnamon essential oil to fabricate nets for use during the fruit-growing period under greenhouse conditions and during storage. Their results showed that the antioxidant capacity was notably higher in tomatoes grown under greenhouse conditions, while the quality was maintained during storage. In addition, Correa-Pacheco et al. [13] packed avocado ‘Hass’ fruit in similar polymeric matrix nets, but added chitosan and pine essential oils. They reported no side effects on the quality of the avocado at the end of the storage period and markedly less infection caused by the fungus *Colletotrichum gloeosporioides*.

Some other studies have described the effects of packaging with plastic materials on the quality and shelf life of bell peppers. For example, Nyanjage et al. [14] found that sweet peppers stored using polyethylene packages had a weight loss of approximately 6.1%, whereas, without packaging, it was 16.2% on the 10th day of storage at ambient temperature. Another study conducted by Abdus Sattar et al. [5] using perforated and non-perforated polyethylene bags reported an extension of the shelf life and quality of hybrid bell peppers by up to 10 days at a controlled temperature of 4 °C using the perforated bags. 

On the other hand, Mexico is the worldwide producer of the cactus stem (*Opuntia ficus-indica* Mill.). According to SIAP [15], the cactus stem production was about 880,000 t per annum, with Morelos being the highest producer state. However, the postharvest waste of this product produced in 2018 amounted to around 40,000 t [16]. These residues, among others, are known to be rich in cellulose, lignin, and hemicellulose [16,17], and according to Scaffaro et al. [18], in conjunction with PLA, these can be used to form a polymeric matrix with excellent mechanical properties, whose stiffness increases up to 135%, with a favorable future in terms of becoming a biodegradable material.

Currently, research on the effect of biodegradable packaging such as nets on the storage life of agricultural products is scarce. Consequently, the objective of this research was to evaluate the effect of nets made of PLA, PBAT, and cactus stem flour obtained from residues on the quality, the bioactive compounds, the nutritional content, and the presence of spoilage microorganisms in ‘California Wonder’ bell peppers of four different colors, during controlled (8 °C) and ambient temperature (30 °C) conditions for 21 and 4 days, respectively.

## 2. Materials and Methods

### 2.1. Fiber Elaboration

The methodology for fiber elaboration was proposed by Black-Solis et al. [12] and Correa-Pacheco et al. [13]. The fibers were extruded from a mixture of two biodegradable polymers: PLA (IngeoTM Biopolymer 7001D, NatureWorks, LLC, Blair, NE, USA) and PBAT (Ecoflex^®^ F Blend C1200, BASF, Mexico City, Mexico) in a 60/40 ratio (PLA/PBAT), and cactus stem flour at 3% using canola oil (Valley Foods^®^, Michoacán, Mexico) at 4% as a plasticizer. For the extrusion, a twin-screw extruder (Process 11, Thermo Scientific™, Waltham, MA, USA) was used with a temperature profile of 160/160/170/180/190/190/160 °C. The fibers were then cooled in water. The 60/40 pellets were dried at 60 °C for 24 h in a conventional oven prior to extrusion. A peristaltic pump (MasterFlex C/L, Cole-Parmer, Vernon Hills, IL, USA) was used for the addition of the cactus stem flour to the second port of the extruder.

### 2.2. Net Bags Elaboration

For knitting the nets, manual circular looms with a diameter of 10 cm and 30 weaving points were used. These allowed obtaining nets with the exact diameter to fit the XL size of the peppers.

### 2.3. Horticultural Material

The bell peppers were cv. California Wonder of four different colors: red, yellow, orange, and green. They were obtained from a greenhouse located in the state of Morelos, Mexico. The fruit was at the commercial ripening stage (100% development color), size XL, and had no defects.

The treatments applied were as follows: T1 = control without nets, T2 = PLA 60%/PBAT 40% nets, T3 = PLA 60%/PBAT 40%/cactus stem flour 3% nets, and T4 = commercial nets. Three peppers were randomly placed in each net and stored at 8 ± 2 °C (controlled temperature) for 21 days and additionally for 4 days at 30 ± 2 °C (ambient temperature). Overall, for each variable, evaluations were taken on days 0, 7, 14, and 21 at the controlled temperature and only on day 25 at room temperature.

### 2.4. Effect of Nets on the Quality and Ripening of the Bell Peppers

#### 2.4.1. Color

The color was measured using a colorimeter CR-10 (Konica-Minolta, Baking Meter BC-10, Japan) obtaining the values of *L** *a**, and *b**. The measurements were taken on two sides of the fruit at the middle point of each one, and values were averaged and reported as
$\Delta E=\sqrt{{\left(L1-L2\right)}^{2}}+{\left(a1-a2\right)}^{2}+{\left(b1-b2\right)}^{2}$. This represents the difference between the initial and final values of *L** (luminosity), *a** (red-green), and *b** (blue-yellow).

#### 2.4.2. Weight Loss

The weight loss variable was calculated as the weight difference between the initial and final days of storage using a balance (Ohaus CS200, Parsippany, NJ, USA). The results were expressed as percentages (%).

#### 2.4.3. Firmness

The firmness was determined at two points in the middle of the fruit using a penetrometer (TR 53,205, Turoni Forli, Italy). The results were averaged and expressed in Newtons (N).

#### 2.4.4. Total Soluble Solids (TSS)

The TSS was obtained using a refractometer (Atago N-1E, Fukaya-shi, Japan), and the results were reported in ºBrix.

#### 2.4.5. Titratable Acidity (TA)

The methodology described for The Association of Official Agricultural Chemists (AOAC) [19] was followed to obtain the titratable acidity. Twenty grams of bell pepper was homogenized with 100 mL of distilled water in a blender (Osterized 2000, Mexico City, Mexico) and titrated with NaOH at 1.0 until a pH of 8.3 was reached. The results were expressed as a percentage (%) based on citric acid.

#### 2.4.6. Respiration Rate

As the nets were not tightly closed, the experimental units of three fruits (red, orange, yellow, or green) were each placed in a sealed jar (vol. 4.0 L) and kept at room temperature (25 ± 2 °C) for 2 h. Analysis of this variable was not carried out in terms of color. The CO_2_ level was determined using gas chromatography (GC, model 7890B, Agilent Technologies, CA, USA). The carrier gas was helium at a flow rate of 10 mL/min. Two detectors, a detector de ionización de llama (FID) and a thermal conductivity detector (TCD) and two HP-PLOT/Q and CP MOLSIEVE 5 A columns (Agilent Technologies, Santa Clara, CA, USA) were used. The oven temperature was 80 °C, and those of the injector and the FID and TCD detectors for CO_2_ were 220 °C, 300 °C, and 250 °C, respectively. The injector was used in the split mode (1:10). The respiration rate was expressed in mL CO_2_ kg^−1^ h^−1^ [12].

### 2.5. Effect of Nets on the Bioactive Compounds Content of the Bell Peppers

The biochemical assays carried out according to the bell pepper color are shown in Table 1.

#### 2.5.1. Carotenoids

To determine the carotenoid content, the methodology proposed by Rodriguez-Amaya [20] was followed. One g of pepper bell (red, orange, and yellow) was macerated with 5 mL of a solution of hexane, ethanol, and acetone (50:25:25). The sample was centrifuged (Prism C2500 centrifuge, Labnet, NJ, USA) at 4293× *g* for 10 min. Subsequently, the hexane phase was recovered and adjusted to 10 mL with pure hexane to measure its absorbance at 450 nm in the spectrophotometer (Genesys 10S UV-VIS, Thermo Fisher Scientific, Waltham, MA, USA). This determination was only evaluated in the case of orange and yellow bell peppers. Data were expressed in µg g FW^−1^.

#### 2.5.2. Anthocyanins

The anthocyanin content was determined only in the case of red peppers following the methodology of Figueroa et al. [21] and Kou et al. [22]. One g of tissue was weighed and macerated with 10 mL of a solution of HCl (35%), distilled water, and methanol (1:49:50). The sample was centrifuged (Prism C2500, centrifuge, Labnet, NJ, USA) at 8000 rpm for 10 min after the supernatant was recovered and adjusted to 10 mL with pure hexane to measure its absorbance at 520 nm using a spectrophotometer (Genesys 10S UV-VIS, Thermo Fisher Scientific, Waltham, MA, USA). The final values were expressed in mg kg^−1^.

#### 2.5.3. Chlorophyll

This determination was evaluated following the methodology reported by Istúriz-Zapata [23], and only for green bell peppers. For this, 10 mg of the epidermis was taken, cut into small pieces, placed in a 3 mL solution of 80% *v*/*v* acetone, incubated for 24 h, and macerated. The homogenate was filtered using Whatman No. 2 filter paper. The filtrate was made up to 10 mL with acetone 80% *v*/*v*, and the absorbance was measured at 645 nm and 663 nm (Genesys 10S UV-VIS, Thermo Fisher Scientific, Waltham, MA, USA). The results were reported as mg of chlorophyll per gram of tissue.

#### 2.5.4. Phenols

Total phenol content was determined using the method proposed by Singleton and Rossi [24]. This consisted of macerating 1 g of bell pepper in 5 mL of 80% methanol. The mixture was centrifuged at 3448× *g* for 10 min, and the supernatant was recovered. Subsequently, 20 μL of the supernatant, 250 μL of Folin–Ciocalteu reagent (Hycel Chemical Reagents, Mexico City, Mexico), and 750 μL of 20% Na_2_CO_3_ (Fermont, Mexico City, Mexico) were mixed and adjusted to 5 mL with distilled water. The mixture was incubated in the dark at 25 ± 3 °C for 2 h, and the absorbance at 760 nm was measured using a spectrometer (Genesys 10S UV-VIS). The results were reported as mg of gallic acid g^−1^ FW.

#### 2.5.5. Flavonoids

The quantification of flavonoids was carried out using the methodology proposed by Chougui et al. [25], which consisted of macerating 0.5 g of pepper with 5 mL of 80% methanol. The solution was centrifuged (Prism C2500, centrifuge, Labnet, NJ, USA) at 8000 rpm for 10 min to recover the supernatant. Then, 1 mL of AlCl_3_ was mixed with 1 mL of the sample and incubated in the dark for 30 min. The absorbance was measured at 430 nm using a spectrophotometer (Thermo Fisher Scientific, MA, USA), and the results were reported as 100 g^−1^ fresh weight quercetin mg. A standard quercetin curve was used for the quantification in 10 to 60 mg quercetin.

### 2.6. Effect of Nets on the Nutritional Content of the Bell Peppers

#### Vitamin C

Ascorbic acid content was determined by spectrophotometry according to the method proposed by Rosales and Arias [26], with some modifications. One gram of tissue was weighed from each fruit to give a total of 5 g, to which a 4% oxalic acid solution was subsequently added to facilitate its grinding in a mortar. The samples were macerated and filtered. Then, 250 μL of green bell pepper was blended with 2250 μL of dichlorophenolindophenol. This was kept in the dark for 15 s, and subsequently, its absorbance at 520 nm was evaluated. To quantify the ascorbic acid in the sample, the following two solutions were used as negative controls: (1) 0.250 mL of oxalic and 2.250 mL of dichlorophenolindophenol, and (2) 2.250 mL of water and 0.250 mL of sample. In addition, a standard curve of ascorbic acid was prepared at concentrations ranging from 1 to 6 mg.

### 2.7. Effect of Nets on the Growth of Spoilage Microorganisms in the Bell Peppers

#### Microbiological Analysis

The microbiological analysis was carried out following the methodology of Rives-Castillo et al. [17]. For this, an aqueous solution of peptone (1 g of casein peptone + 8.5 g of sodium chloride + 1 L of distilled water) was prepared to rinse the fruit pool per treatment under sterile conditions. A total of 500 μL of the water obtained was taken, and a dilution was made (10^−6^). This was poured into Petri dishes with potato dextrose agar (PDA) to isolate the fungi and yeast or soy trypticase agar (AST) for bacterial analysis (BIOXON, Mexico City, México), and incubated for 24 h at 30 °C. The microbiological analysis was performed at the end of the storage period, corresponding to 8 °C for 21 days and at 30 °C for 4 days conditions. The presence of spoilage microorganisms was reported in colony-forming units (CFU) at 10^−6^ dilutions.

### 2.8. Statistical Analysis

Treatments were arranged in a completely randomized design. All analyses were carried out in triplicates (three nets per treatment, containing three fruits in each). The data were analyzed with an ANOVA and a Tukey’s test (*p* < 0.05) using the SigmaPlot 12.5 software (Systat Software Inc., Palo Alto, CA, USA, 2011). The standard deviations of the media were also included.

## 3. Results and Discussion

### 3.1. Effect of Nets on the Quality and Ripening Behavior of the Bell Peppers

#### 3.1.1. Color

In this study, regardless of the bell pepper color and type of net, in general, the development of color under controlled and ambient temperatures was very similar (Figure 1a), showing no statistical differences over time. However, differences in ΔE were observed due to the effect of the color of the fruit, with the orange and yellow bell peppers reaching the highest values (ΔE = 50). These differences could be due to the diversity of pigments that prevail in each of them, such as carotenoids and capsanthin in red bell peppers, and chlorophylls in green bell peppers.

Contrary to these findings, Frans et al. [27] indicated that ΔE color was different between the red and yellow bell peppers when they were stored in perforated low-density polyethylene (LDPE) bags at 20 °C for 14 days. In addition, the color of the red variety was more intense and stable when stored under modified packaging atmosphere (MAP) conditions compared to the oxidative brown color observed in the control fruit, whereas, for the yellow peppers, the color changes were in accordance with the ripening process. In addition, Mangaraj et al. [28] reported a decrease in color by up to 50% in the case of green bell peppers stored at 8 and 25 °C for 12 days and packed in bag-based PLA/corn starch 20%. The authors explained that this might be due to ‘pigment degradation during the ripening and browning by polyphenol oxidase’.

#### 3.1.2. Weight Loss

With respect to weight loss, bell peppers with and without nets gradually lost weight during the whole storage period at controlled and ambient temperatures; however, the final values were not statistically different among treatments, the color of the peppers, and storage temperature. Generally, the weight loss ranged from 13 to 25% at 8 °C and from 13 to 33% at 30 °C. At the end of the storage period, the lowest weight loss (of approximately 20%) was observed for the yellow and green bell peppers stored without nets (T1) and with PLA 60%/PBAT 40%/cactus stem flour 3% nets, respectively (T3) (Figure 1b). 

Authors such as Mangaraj et al. [28], Chitavathri et al. [29], and Lama et al. [30] among others highlighted that unpacked bell peppers and green chilies usually showed the highest weight loss compared to those stored in plastic and polymeric packaging films at a controlled temperature. In this study, in the case of the unpacked yellow and green bell peppers, the lowest weight effect was probably due to the initial ripening stage.

#### 3.1.3. Firmness

For this variable, there were statistically significant differences (*p* < 0.05) among treatments, sampling days (0, 7, 14, 21, and 25), and storage temperatures (8 and 30 °C) (Figure 2). In general, the highest firmness loss was observed in the case of peppers stored without nets and in commercial nets. Regardless of the peppers’ color, there was a clear loss of firmness as the storage time increased, with a marked loss during the four days at 30 °C. In this case, the corresponding final values ranged from approximately 15 N to 28 N for red, yellow, and green bell peppers, whereas the orange bell pepper had a firmness of up to 35 N.

Similar behavior was reported by Chitavathri et al. [29]. In that study, there was a dramatic decrease in the firmness of green chilies as the period of storage at 8 °C increased. Similarly, Sattar et al. [31] tested sealed polyethylene bags against perforated polyethylene bags on hybrid bell peppers at 4 °C and 27 °C, resulting in less firmness loss in those kept in perforated bags than those kept in sealed bags, with a tendency for higher weight loss as the temperature increased. In addition, Mangaraj et al. [28] reported that the use of bags based on PLA-corn (10% + 20%) maintained a higher firmness compared to unpackaged bell peppers, extending their shelf life by up to 9 days at 30 °C.

#### 3.1.4. Total Soluble Solids

In terms of this variable, there were significant differences (*p* < 0.05) among treatments, storage days, and temperatures (Table 2). Overall, during the entire storage period at both temperatures, the lowest TSS content corresponded to the green peppers, whereas the highest values were found in red, orange, and yellow peppers. In addition, in the last three colors, regardless of the treatment, including the control, the TSS slightly increased during storage. No pattern was observed with regard to the use of nets. Independent of the fruit color, the final average values of TSS at controlled temperature were between 5.0 and 11.0, whereas at the ambient temperature, they were found to range from 6.0 to 11.0 TSS.

In agreement with these results, Abdus et al. [5] and Frans et al. (2021) [27] reported minimum changes in this variable in terms of the bell pepper variety, storage temperatures of 4, 6.5, and 25 °C, types of packing (perforated, non-perforated, and without packing), and preharvest conditions.

#### 3.1.5. Titratable Acidity

With respect to the TA, there were no statistical differences among treatments, storage days, and temperatures; however, the difference occurred in the variable color. In this case, the green bell peppers had the lowest TA values (from 0.03% to 1%). Mangaraj et al. [28] obtained similar values compared to ours of approximately 0.20% in yellow peppers stored at 8 °C. Due to the fact that peppers are classified as a non-climacteric fruit, TSS and TA showed little change.

#### 3.1.6. Respiration Rate

With respect to the respiration rate expressed as CO_2_ production, overall, it increased during the entire storage period, but without any significant statistical differences among treatments (Figure 3). The CO_2_ production of bell peppers was in a range of approximately 2.4 to 3.0 mL of CO_2_ kg^−1^ h^−1^ at the controlled temperature, whereas CO_2_ production in all treatments clearly increased at ambient temperature from 2.4 to 4.9 mL of CO_2_ kg^−1^ h^−1^. The lowest CO_2_ production corresponded to the bell peppers kept in commercial nets and non-bagged, with a marked difference at the low storage temperature compared to the ambient temperature.

The same effect associated with the temperature factor was reported by Mangaraj et al. [28] in the case of capsicums stored in film-based PLA/corn starch. There was an increase in CO_2_ concentration as temperature increased (10 to 30 °C) during the 60 h of the sampling period. In another investigation by Lwin et al. [31], there was an increase in CO_2_ production in the case of unpacked bell pepper cultivars compared to those kept in perforated and non-perforated MAP films stored at 22 °C for up to three weeks, an outcome that was not observed in our research.

### 3.2. Effect of Nets on the Bioactive Compounds of the Bell Peppers

#### 3.2.1. Carotenoids

In the case of this variable, the results showed statistical differences (*p* < 0.05) among treatments and sampling days for each bell pepper color. In general, the lowest carotenoid content during the entire storage period was in the case of yellow peppers, followed by orange and red bell peppers. With regard to the tested nets, there was not a clear pattern associated with their effect on the bell peppers. The differences were more probably in accordance with their harvest maturity. The corresponding ranges were 30 to 65 µg g FW^−1^ for the red color (Figure 4a), 5.0 to 70 µg g FW^−1^ for orange (Figure 4b), and 5 to 38 µg g FW^−1^ for yellow (Figure 4c).

Coinciding with our results, Pugliese et al. [32] noted high values of β-carotene in different cultivars of red bell peppers. Abdus et al. [5] reported a better quality of sweet bell peppers when stored at 4 °C in perforated polyethylene poly bags compared to non-perforated packaging. In the case of β-carotene, they reported a slight increase during the 20 days of storage, but higher values at 4 °C (44.5 µg 100 g^−1^) than at ambient temperature (42.5 µg 100 g^−1^).

#### 3.2.2. Anthocyanins

For this variable, there were significant differences (*p* < 0.05) from day 14 to 25 among the treatments. The lowest values were always associated with the peppers kept in commercial nets at both temperatures, with the following final corresponding values: approximately 26 mg kg^−1^ at 4 °C after 21 days and 32 mg kg^−1^ at 30 °C for 4 days (Figure 5). The highest content corresponded to the fruits kept in PLA 60%/PBAT 40%/cactus stem flour 3%, with a value of 40 mg kg^−1^ at 30 °C.

According to Niu et al. [33], the synthesis of anthocyanins in plants is strongly associated with various stress factors including, among others, sun radiation, temperature, plant organ, and age. For example, low temperatures increased the anthocyanin content in the leaves of evergreen plant species, whereas high temperatures decreased the anthocyanins of pear peel. In our study, except for T3 and the storage temperature, the concentration of anthocyanins was the lowest in the remaining treatments. Studies by Mohaven et al. [34] on Sangiovese berries reported the lowest values of anthocyanins in the skin of fruit when grown at 29 °C compared to 20 °C, but at harvest, the anthocyanin content was very similar.

#### 3.2.3. Chlorophyll

For this variable, in each storage period, there were significant differences among treatments (*p* < 0.05). Except for the fruits kept in the net-based PLA 60%/PBAT 40%, the chlorophyll content increased throughout the two storage periods, showing the highest values of approximately 60 mg L^−1^ in the peppers stored in PLA 60%/PBAT 40%/cactus stem residues 3% at 30 °C (Figure 6). At this temperature, the chlorophyll content of the peppers was the highest for most treatments.

In another study reported by Chitravathi et al. [29], the packaging type slightly influenced the chlorophyll amount of green bell peppers stored at 8 °C. In that research, it was the storage period rather than the packing materials that had the most influence, with a notable decrease as the storage of the green bell peppers increased up to 28 days.

This may be related to the increase in TSS reported by García-Jiménez et al. [35]. They highlighted that the increase in chlorophyll during storage was mainly due to the increase in soluble sugars; however, in our study, the increase in the TSS of green bell peppers was minimal. Another explanation could be the inhibition of chlorophyll degradation due to the reduction in the activities of peroxidase that degrade chlorophyll [36].

#### 3.2.4. Phenols

In general, the phenol content of the evaluated bell peppers increased during the 25-day storage (Table 2). Statistical differences (*p* < 0.05) were associated with treatment, group of pepper color, and days of storage. Independent of color, overall, the phenol content was higher in the samples stored in the net-based PLA 60%/PBAT 40%/cactus stem flour 3%, probably because the nets were a stress factor that stimulated the synthesis of phenols. After 21 days, the values ranged from 1.1 to 6.2 mg of gallic acid g^−1^, whereas on day 25, the values ranged from 0.05 to 5.3 mg of gallic acid g^−1^. Regardless of the treatments, the green peppers showed the lowest phenol content during the storage period.

In agreement with our results, Chitravati et al. [29] and Barzegar et al. [2] also reported an overall increase in phenol content during storage time in green bell peppers. For example, Chitravati et al. [29] reported slight differences depending on the packing material, whereas Devgan et al. [37] outlined an average value of total phenol content in yellow peppers of 1.5 mg/100 g. In that study, the values of the total phenols in the samples showed an increasing trend during storage for samples stored in low-density polyethylene (LDPE) without an oxygen absorber, whereas, for samples stored with an oxygen absorber, the total phenols first increased and then decreased.

#### 3.2.5. Flavonoids

For this variable, there were statistical differences (*p* < 0.05) regarding the treatment, group of pepper color, and days of storage (Table 3). Although there was no pattern with regard to the effect of the factors on the flavonoid content, in general, with respect to the initial values, the flavonoid content increased during storage. At the end of the storage period, the red and yellow peppers showed the highest average values of approximately 189.9 µg^−1^ quercetin g^−1^, followed by green peppers with a corresponding average value of 133 µg^−1^ quercetin g^−1^.

Chitravathi et al. [29] and Barzegar et al. [2] reported a moderate increase in this bioactive compound during the storage of bell peppers, regardless of the storage temperature. However, different results were obtained by Barbosa et al. [38]. They pointed out that the evolution of the content of flavonoids in bell peppers (green, red, and yellow) was very similar throughout the storage period when ‘Krehalon’ bags were used in a modified atmosphere during 17 days of storage at 5 °C.

### 3.3. Effect of Nets on the Nutritional Content of the Bell Peppers

#### Vitamin C

The amount of vitamin C for each bell pepper color showed significant statistical differences (*p* < 0.05) among treatments and groups of colors during the 21 days of storage at 8 °C. The red peppers (approximately 280 mg/100 g^−1^ ascorbic acid), orange peppers (approximately 260 mg/100 g^−1^ ascorbic acid), and yellow peppers (approximately 320 mg/100 g^−1^ ascorbic acid) had the highest average content of ascorbic acid compared to the green group (90 mg/100 g^−1^ of ascorbic acid) that had the lowest values (Figure 7). In addition, all groups of peppers stored in the net-based PLA 60%/PBAT 40%/cactus stem flour 3% (T3) showed the highest average value of approximately 235 mg/100 g^−1^ ascorbic acid.

Contrary to our results, Frans et al. [27] reported that in four different cultivars of red and yellow bell peppers stored in LDPE bags at 20 °C for 14 days, there were similar concentrations of vitamin C (±1.05 g kg^−1^), concluding that MAP did not influence the vitamin C content in any of the cultivars or with regard to any of the treatments. On the other hand, according to Lama et al. [30], the levels of ascorbic acid in yellow bell peppers were hardly affected by the storage temperatures of 4 °C, 7 °C, and ambient temperature combined with microperforated bags. They reported that there were other factors such as the harvest season that influenced this variable to a greater extent.

Dogan et al. [39] outlined that the ascorbic acid content in pesticide-free and conventionally grown red bell peppers declined progressively during the 45 days of storage at 8 °C, but with differences according to the type of storage. For example, there was a higher level of this nutritional compound when kept under a controlled atmosphere compared to the unpacked, cling film, and MAP treatments.

Another study carried out by Chen et al. [40] mentioned an increase in vitamin C in red and green bell peppers stored in perforated films based on xanthan gum/hydroxypropyl methylcellulose for 8 days at 25 °C, in contrast to peppers stored without packaging, which had an accelerated decline in vitamin C during storage, leading to early senescence.

### 3.4. Effect of Nets on the Growth of Spoilage Microorganisms in the Bell Peppers

#### Microbiological Analysis

The presence of bacteria, fungi, and yeast was statistically different (*p* < 0.01) among the color groups, treatments, and storage periods (Table 4). With respect to the bacterial count, regardless of storage temperature, bell peppers red, orange, and yellow tended to have the lowest microorganism count (from 3 to 20 times less) when stored in the net-based PLA 60%/PBAT 40%/cactus stem flour 3% (T3) compared to bell peppers stored in PLA 60%/PBAT 40% (T2) and without nets (T1). Except for the green peppers, a low count was also observed in the bell peppers stored in T4. With regard to fungi and yeast, the lowest average counts corresponded to the fruits stored at 8 °C in T3, with corresponding average values of 27 CFU. In the bell peppers stored in commercial nets, the final values were 53 CFU at 8 °C and 43 CFU at 30 °C. The orange and green bell peppers stored without nets had the highest fungi and yeast counts (350 CFU and 299 CFU at 8 °C, respectively). Finally, the green pepper packed in commercial nets (T4) showed a considerably higher difference in the bacteria and fungi counts compared to the other treatments. This was probably due to an accelerated ripening process, their antioxidant and nutritional composition, and a major susceptibility compared to other colors.

Due to the nutritional content of fruits and vegetables, they can be infected by a wide range of microorganisms that cause deterioration and diseases [41]. An inhibitory effect of PLA and PBAT on the growth of spoilage microorganisms in polymeric matrices composed has not yet been reported. This opens up the opportunity to further investigate the possible effects that these nets could have on the development of foodborne pathogens.

## 4. Conclusions

The use of PLA 60%/PBAT 40%/cactus stem flour 3%-based nets during the storage of bell peppers, regardless of the color cultivar, generally maintained the same quality as those kept in commercial nets. Occasionally, the nets increased the content of the bioactive compounds and reduced the presence of bacteria, fungi, and yeast. As expected, the storage temperature influenced the ripening of the bell peppers, accelerating this process during the four-day storage period at 30 °C. As a postharvest packaging for bell peppers, the biodegradable packaging based on PLA 60%/PBAT 40%/cactus nopal flour 3% could be considered a viable option for reducing the environmental impact and postharvest loss due to the excessive use of packaging agricultural plastics, in this case, for bell peppers.

## Figures and Tables

**Figure 1 foods-12-02071-f001:**
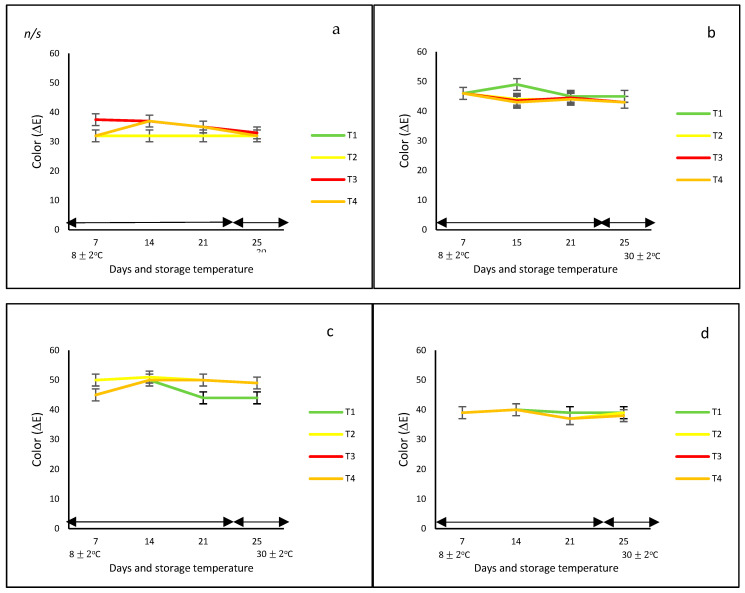
Color change (ΔE) (**a**–**d**) and weight loss (**e**–**h**) in ‘California Wonder’ bell peppers (red **a**,**e**; orange **b**,**f**; yellow **c**,**g**; and green **d**,**h**) stored without nets (control) (T1), PLA 60%/PBAT 40% nets (T2), PLA 60%/PBAT 40%/cactus stem flour 3% nets (T3), and commercial nets (T4) at 8 ± 2 °C for 21 days and at 30 ± 2 °C for 4 days. Bars indicate the standard deviations of the means.

**Figure 2 foods-12-02071-f002:**
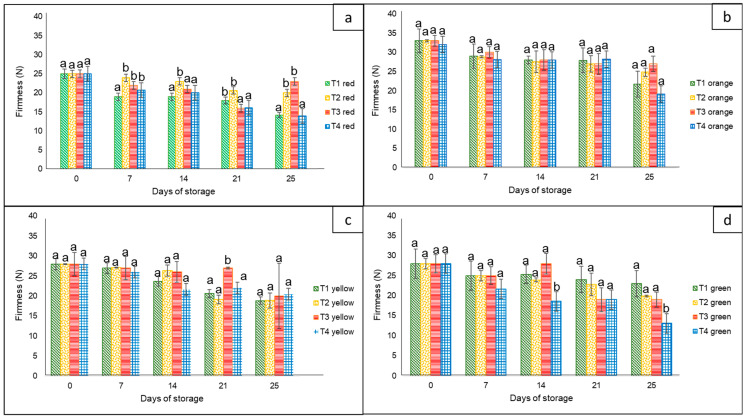
Firmness in ‘California Wonder’ bell peppers red (**a**), orange (**b**), yellow (**c**), and green (**d**), stored without nets (control) (T1), PLA 60%/PBAT 40% nets (T2), PLA 60%/PBAT 40%/cactus stem flour 3% nets (T3), and commercial nets (T4), at 8 and 30 °C for 21 and 4 days, respectively. Different letters represent statistical differences (*p* < 0.05). Bars indicate the standard deviations of the means.

**Figure 3 foods-12-02071-f003:**
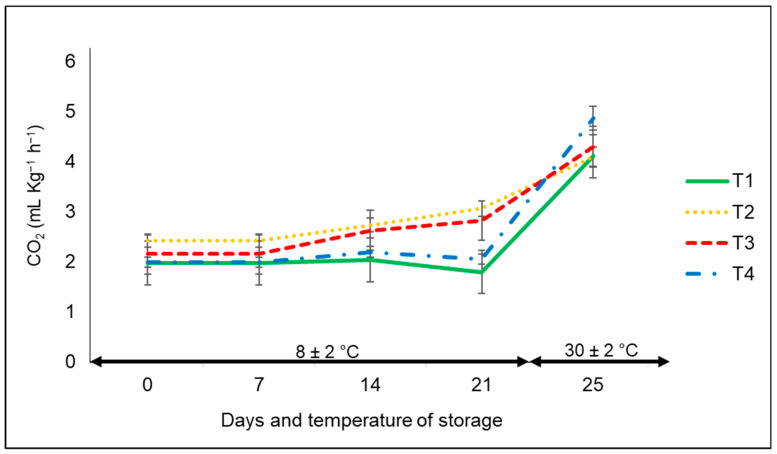
CO_2_ production in ‘California Wonder’ bell peppers stored without nets (control) (T1), PLA 60%/PBAT 40% nets (T2), PLA 60%/PBAT 40%/cactus stem flour 3% nets (T3), and commercial nets (T4) at 8 and 30 °C for 21 and 4 days, respectively. Bars indicate the standard deviations of the means.

**Figure 4 foods-12-02071-f004:**
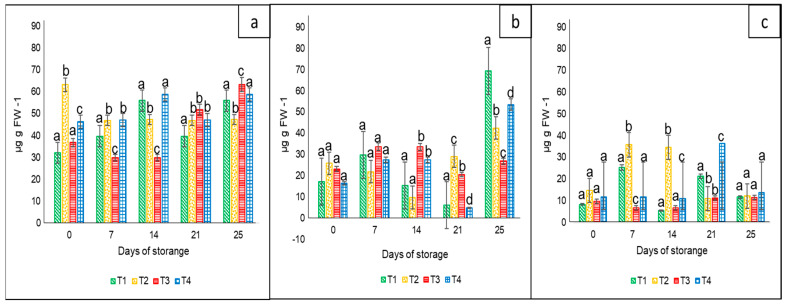
Carotenoid content in ‘California Wonder’ bell peppers red (**a**), orange (**b**), and yellow (**c**) stored without nets (control) (T1), PLA 6%/PBAT 40% (T2), PLA 6%/PBAT 40%/cactus stem flour 3% (T3), and commercial nets (T4) at 8 and 30 °C during given storage days. Different letters represent statistical differences (*p* < 0.05). Bars indicate the standard deviations of the means.

**Figure 5 foods-12-02071-f005:**
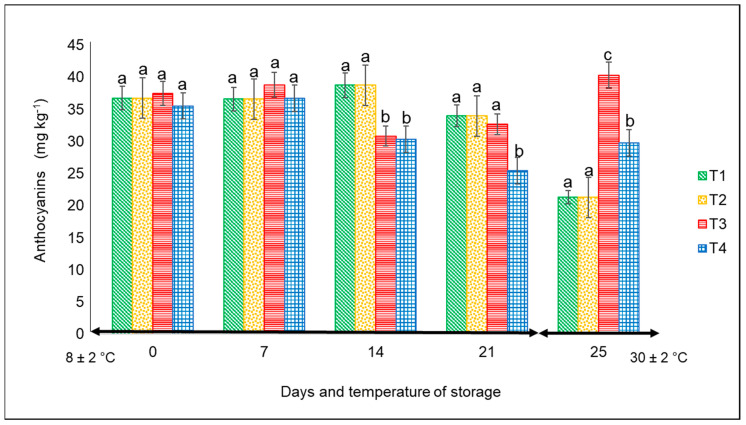
Content of anthocyanins in ‘California Wonder’ red bell peppers stored without nets (control) (T1), PLA 60%/PBAT 40% nets (T2), PLA 60%/PBAT 40%/cactus stem flour 3% nets (T3), and commercial nets (T4) at 8 and 30 °C during given storage days. Different letters represent statistical differences (*p* < 0.05). Bars indicate the standard deviations of the means.

**Figure 6 foods-12-02071-f006:**
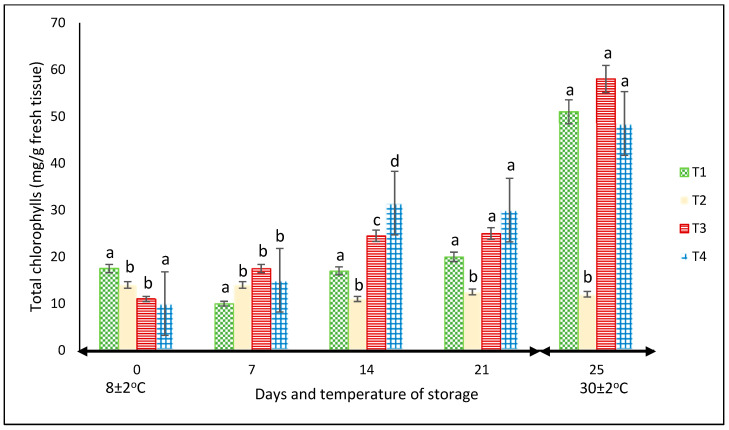
Content of chlorophyll in ‘California Wonder’ green bell peppers stored without nets (control) (T1), PLA 60%/PBAT 40% nets (T2), PLA 60%/PBAT 40%/cactus stem flour 3% nets (T3), and commercial nets (T4) at 8 and 30 °C during given storage days. Different letters represent statistical differences (*p* < 0.05). Bars indicate the standard deviation of the means.

**Figure 7 foods-12-02071-f007:**
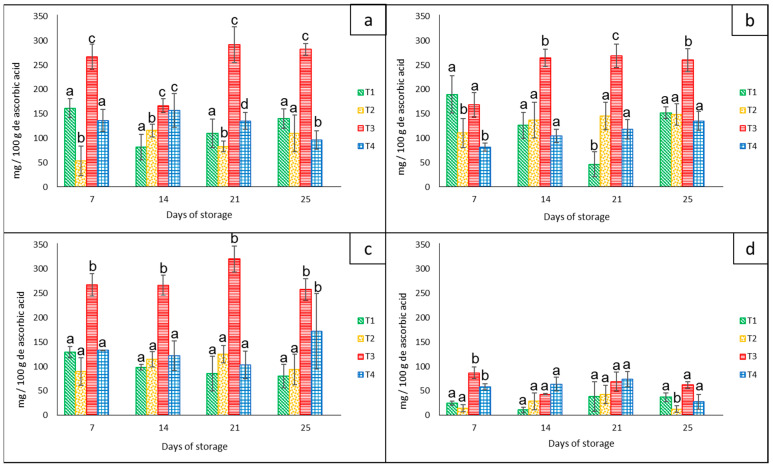
Content of ascorbic acid in ‘California Wonder’ bell peppers red (**a**), orange (**b**), yellow (**c**), and green (**d**) stored without nets (control) (T1), PLA 60%/PBAT 40% nets (T2), PLA 60%/PBAT 40%/cactus stem flour 3% nets (T3), and commercial nets (T4) at 8 and 30 °C during given storage days. Different letters represent statistical differences (*p* < 0.05). Bars indicate the standard deviations of the means.

**Table 1 foods-12-02071-t001:** Biochemical tests evaluated in bell peppers.

Test	Bell Pepper Color
Carotenoids	Red, orange, yellow
Anthocyanins	Red
Chlorophyll	Green
Phenols	Red, orange, yellow, and green
Flavonoids	Red, orange, yellow, and green
Vitamin C	Red, orange, yellow, and green

**Table 2 foods-12-02071-t002:** Content of TSS in ‘California Wonder’ bell peppers (red, yellow, orange, and green) stored with and without nets at 8 and 30 °C during given storage days.

T	Bell Pepper Color	Storage Temperatures			
		8 ± 2 °C			30 ± 2 °C
		Storage Period (Days)			
		7	14	21	25
1	red	8.2 ± 0.7 ^aA^	8.2 ± 1.9 ^aA^	8.2 ± 1.1 ^aA^	8.2 ± 0.1 ^aA^
1	orange	7.8 ± 0.1 ^aA^	7.0 ± 0.03 ^bA^	7.1 ± 2.0 ^bA^	8.3 ± 1.1 ^bA^
1	yellow	7.2 ± 2.0 ^aA^	7.8 ± 0.1 ^bA^	7.8 ± 0.7 ^bA^	8.0 ± 1.9 ^bA^
1	green	5.2 ± 1.1 ^bA^	5.0 ± 0.7 ^aA^	5.0 ± 0.1 ^aA^	6.0 ± 2.0 ^aA^
2	red	8.2 ± 0.2 ^aA^	8.2 ± 1.1 ^aA^	8.0 ± 0.9 ^aA^	8.8 ± 0.2 ^bA^
2	orange	7.5 ± 0.1 ^aB^	7.8 ± 0.05 ^bC^	7.8 ± 1.1 ^bA^	7.8 ± 0.1 ^bA^
2	yellow	8.9 ± 2.0 ^bB^	8.2 ± 0.1 ^aB^	8.8 ± 0.1 ^bA^	9.0 ± 2.0 ^bA^
2	green	5.2 ± 1.1 ^aA^	6.2 ± 2.0 ^aA^	5.2 ± 1.1 ^aA^	5.0 ± 0.1 ^aA^
3	red	8.2 ± 0.1 ª^aA^	9.0 ± 1.2 ^aA^	10.2 ± 0.9 ^aA^	11.0 ± 0.2 ^bA^
3	orange	8.2 ± 2.0 ^aA^	8.2 ± 0.05 ^aA^	8.1 ± 1.1 ^bA^	10.0 ± 1.9 ^aA^
3	yellow	9.2 ± 1.1 ª^aA^	9.2 ± 0.1 ^aA^	11.0 ± 0.1 ^bA^	10.0 ± 0.2 ^aA^
3	green	5.0 ± 0.1 ª^aA^	5.2 ± 0.2 ^aA^	5.2 ± 0.9 ^aA^	6.3 ± 0.1 ^aA^
4	red	8.4 ± 1.9 ^aA^	8.5 ± 2.0 ^bB^	8.5 ± 1.1 ^bA^	8.0 ± 0.1 ^aA^
4	orange	8.0 ± 1.1 ^aB^	8.0 ± 1.1 ^aB^	8.0 ± 1.9 ^aA^	8.0 ± 0.7 ^aA^
4	yellow	7.0 ± 0.03 ^aA^	7.4 ± 1.1 ^aB^	8.0 ± 0.1 ^bA^	8.0 ± 1.1 ^bA^
4	green	5.5 ± 0.1 ^aA^	5.8 ± 0.2 ^aA^	5.0 ± 2.0 ^aA^	6.5 ± 1.9 ^aA^

T1 = without nets (control), T2 = PLA 60%/PBAT 40% nets, T3 = PLA 60%/PBAT 40%/flour cactus stem 3% nets, and T4 = commercial nets. Different letters represent statistical differences according to ANOVA and Tukey’s test (*p* < 0.05). Lower cases correspond to differences among storage days, and upper cases to differences among treatments. Data represent means ± standard deviations. Initial values of peppers were as follows: red = 8.0, orange = 7.5, yellow = 7.2, and green = 5.0.

**Table 3 foods-12-02071-t003:** Phenol and flavonoid content in ‘California Wonder’ bell peppers (red, yellow, orange, and green) stored with and without nets at 8 and 30 °C during given storage days.

Phenolic Content (mg of Gallic Acid g^−1^) Storage Temperatures and Days					
	Bell Pepper	8 ± 2 °C			30 ± 2 °C
T	Color	7	14	21	25
1	red	2.4 ± 2.2 ^aA^	3.2 ± 1.5 ^aA^	5.0 ± 1.4 ^bA^	5.0 ± 1.1 ^bA^
1	orange	2.7 ± 1.3 ^bA^	3.1 ± 0.9 ^bA^	4.7 ± 0.3 ^cA^	5.3 ± 0.2 ^dA^
1	yellow	3.0 ± 1.2 ^aA^	3.5 ± 1.7 ^aA^	5.4 ± 0.7 ^bA^	5.3 ± 0.7 ^bA^
1	green	0.6 ± 0.3 ^aA^	3.9 ± 1.9 ^bA^	1.1 ± 0.9 ^aA^	3.0 ± 2.5 ^bA^
2	red	3.2 ± 1.7 ^aA^	3.1 ± 1.2 ^aA^	5.2 ± 1.5 ^bA^	4.6 ± 0.9 ^bA^
2	orange	3.8 ± 1.7 ^bB^	4.1 ± 0.7 ^cA^	5.5 ± 0.8 ^dA^	5.4 ± 0.7 ^dB^
2	yellow	1.2 ± 2.6 ^bB^	2.0 ± 1.2 ^bA^	5.3 ± 0.8 ^cC^	5.1 ± 0.9 ^cB^
2	green	2.2 ± 2.1 ^bB^	2.0 ± 1.5 ^bA^	3.3 ± 2.8 ^bB^	2.8 ± 1.7 ^bA^
3	red	3.0 ± 1.3 ^aA^	3.1 ± 0.5 ^aA^	6.2 ± 0.8 ^cA^	4.7 ± 0.8 ^bB^
3	orange	2.9 ± 0.6 ^aA^	2.4 ± 1.3 ^aA^	2.8 ± 0.7 ^aB^	3.5 ± 1.2 ^bA^
3	yellow	1.0 ± 2.0 ^bB^	0.8 ± 0.6 ^bC^	3.2 ± 0.7 ^bB^	1.6 ± 0.9 ^bA^
3	green	0.7 ± 0.2 ^bB^	0.4 ± 0.7 ^bC^	1.2 ± 0.2 ^dA^	0.1 ± 1.0 ^cD^
4	red	2.2 ± 2.9 ^aA^	3.1 ± 0.8 ^aA^	4.8 ± 1.0 ^bA^	5.0 ± 0.5 ^bA^
4	orange	1.4 ± 0.7 ^aA^	4.1 ± 2.0 ^aA^	4.9 ± 1.0 ^bA^	4.7 ± 1.1 ^bB^
4	yellow	1.7 ± 1.5 ^aB^	2.1 ± 1.4 ^bA^	3.0 ± 2.0 ^aB^	4.2 ± 1.1 ^bB^
4	green	2.5 ± 2.4 ^aB^	3.9 ± 1.8 ^bA^	1.5 ± 1.5 ^bA^	1.8 ± 2.3 ^bA^
Flavonoids (μg quercetin g^−1^)					
1	red	191.7 ± 0.1 ^bA^	149.8 ± 0.1 ^cA^	209.4 ± 0.1 ^dA^	190.0 ± 0.1 ^eA^
1	orange	136.1 ± 0.1 ^bA^	93.2 ± 0.1 ^cA^	93.9 ± 0.1 ^dA^	73.2 ± 0.1 ^eA^
1	yellow	177.0 ± 0.1 ^bA^	198.6 ± 0.1 ^cA^	160.0 ± 0.1 ^dA^	188.2 ± 0.1 ^eA^
1	green	233.8 ± 0.1 ^cA^	82.3 ± 0.1 ^cA^	87.1 ± 0.1 ^dA^	72.3 ± 0.1 ^eA^
2	red	165.5 ± 0.1 ^bB^	181.0 ± 0.1 ^cB^	184.0 ± 01 ^dB^	146.3 ± 0.1 ^eB^
2	orange	86.3 ± 0.1 ^bB^	57.0 ± 0.1 ^cB^	63.0 ± 0.1 ^dB^	111.7 ± 0.1 ^eB^
2	yellow	109.7 ± 0.1 ^bB^	94.6 ± 0.1 ^cB^	94.7 ± 0.1 ^dB^	177.2 ± 0.1 ^eB^
2	green	98.1 ± 0.1 ^bB^	122.2 ± 0.1 ^cB^	143.0 ± 0.1 ^dB^	103.1 ± 0.1 ^eB^
3	red	90.4 ± 0.1 ^bB^	161.5 ± 0.1 ^cB^	210.7 ± 0.1 ^dB^	106.6 ± 0.2 ^eB^
3	orange	76.0 ± 0.1 ^bD^	92.6 ± 0.1 ^cA^	171.4 ± 0.1 ^dA^	87.0 ± 0.2 ^eA^
3	yellow	113.8 ± 0.1 ^bC^	196.8 ± 0.1 ^cB^	277.1 ± 0.1 ^dB^	140.1 ± 0.3 ^eB^
3	green	31.1 ± 0.1 ^bE^	64.6 ± 0.1 ^cA^	67.4 ± 0.1 ^dB^	34.4 ± 0.1 ^eD^
4	red	189.0 ± 0.1 ^bC^	153.5 ± 0.1 ^cC^	141.1 ± 0.1 ^dC^	155.3 ± 0.1 ^eC^
4	orange	117.1 ± 0.1 ^bC^	97.7 ± 1.1 ^cC^	89.0 ± 0.1 ^aC^	96.8 ± 0.1 ^eC^
4	yellow	110.1 ± 0.1 ^bC^	101.8 ± 0.1 ^cC^	108.2 ± 0.1 ^dC^	107.2 ± 0.1 ^eC^
4	green	112.3 ± 0.1 ^bC^	130.3 ± 0.1 ^cC^	110.1 ± 0.1 ^dC^	133.2 ± 0.1 ^eC^

T = treatment, T1 = without nets (control), T2 = PLA 60%/PBAT 40% nets, T3 = PLA 60%/PBAT 40%/cactus stem flour 3%, and T4 = commercial nets. Different letters represent statistical differences according to ANOVA and Tukey’s test (*p* < 0.05). Lower cases correspond to differences among storage days, and upper cases to differences among treatments. Values represent means ± standard deviations. Initial phenolics values: red = 2.0, orange = 1.4, yellow = 3.0, green = 0.4 mg of gallic acid g^−1^. Initial flavonoids values: red = 160.1, orange = 124.0, yellow = 103.0, green = 73.4 µg^−1^ quercetin g^−1^.

**Table 4 foods-12-02071-t004:** Presence of spoilage microorganisms on ‘California Wonder’ bell peppers (red, orange, yellow, and green) kept with and without nets, after 21 days at 8 °C and 4 days of storage at 30 °C, respectively.

		Bacteria (CFU)		Fungi and Yeast (CFU)	
		8 ± 2 °C	30 ± 2 °C	8 ± 2 °C	30 ± 2 °C
Treatment	Color	21 Days	4 Days	21 Days	4 Days
1	Red	46 ± 0.3	603 ± 21.6	26 ± 5.6	87 ± 21.7
1	Yellow	61 ± 3.0	557 ± 18.4	93 ± 3.7	72 ± 22.0
1	Orange	244 ± 11.2	563 ± 16.7	350 ± 15.4	83 ± 12.1
1	Green	442 ± 11.9	469 ± 9.5	299 ± 12.6	145 ± 10.6
2	Red	274 ± 8.4	*	237 ± 6.9	478 ± 10.2
2	Yellow	186 ± 8.7	*	172 ± 12.8	369 ± 10.2
2	Orange	233 ± 6.5	*	41 ± 8.1	444 ± 21.8
2	Green	657 ± 6.2	*	82 ± 8.8	279 ± 10.5
3	Red	13 ± 4.9	29 ± 4.7	41 ± 0.2	141 ± 2.0
3	Yellow	13 ± 4.2	36 ± 2.2	44 ± 0.6	202 ± 3.0
3	Orange	14 ± 1.0	35 ± 9.1	11 ± 2.1	249 ± 9.4
3	Green	16 ± 0.2	41 ± 1.4	13 ± 2.0	169 ± 1.9
4	Red	19 ± 5.3	26 ± 8.1	23 ± 0.4	82 ± 7.9
4	Yellow	25 ± 5.7	14 ± 8.6	31 ± 0.8	82 ± 0.1
4	Orange	38 ± 6.8	40 ± 10.7	28 ± 1.5	93 ± 0.1
4	Green	23 ± 9.4	146 ± 14.7	132 ± 9.4	87 ± 6.1

* Uncountable. T1 = without nets (control), T2 = PLA 60%/PBAT 40%, T3 = PLA 60%/PBAT 40%/cactus stem flour 3%, and T4 = commercial nets. Values represent means ± standard deviations.

## Data Availability

The data are available from the corresponding authors.

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
