# Peer review of "The Effect of Netting Bags on the Postharvest Quality, Bioactive and Nutritional Compounds, and the Spoilage Microorganisms Content of Bell Peppers"

_foods, 2023, doi:10.3390/foods12102071_

Round 1

Reviewer 1 Report

Dear Authors,

After reading the manuscript entitled "The effect of netting bags on the postharvest quality, bioactive and nutritional compounds, and the spoilage microorganisms content of bell peppers".

Minor corrections in the manuscript text must be performed to increase its quality:

line 39 is:..aspects water..., should be:.. aspects as water..

line 43 is:.. packaging.., should be:.. packagings…

line 47 is:.. are.., - remove

line 50 is:.. centered..., should be:.. centred..

line 55 is:.. of.., - remove

line 65 is:.. plastics..., should be:. plastic..

line 56 is:.. its.., should be:. their..

lines 73, 75 is:.. ton..., should be:..tons..

line 250 Figure 1 – no bars are visible in Figure 1

Figure 1b – change the scale on the Y-axis from 0 to 40; the lines will be more visible.

lines 236, 237 is:.. 13.0..., should be:..13..

line 257 is:.. to 20 N... – are you sure? See Figure 2b after 25 days T1, T2, T3 the values are higher than 20 N

line 295 is:.. were very similar. Data shown an average range from 0.06 to 0.15%...., should be: ...were very similar and ranged from 0.06 to 0.15%.

line 367 is:.. 60 mg/L..., should be:. 60 mg L-1

Author Response

Reviewer 1

line 39 is:..aspects water..., should be:.. aspects as water..

Correction was done.

line 43 is:.. packaging.., should be:.. packagings…

Correction was done.

line 47 is:.. are.., - remove

Correction was done.

line 50 is:.. centered..., should be:.. centred..

Correction was done.

line 55 is:.. of.., - remove

Correction was done.

line 65 is:.. plastics..., should be:. plastic..

Correction was done.

line 56 is:.. its.., should be:. their..

Correction was done.

lines 73, 75 is:.. ton..., should be:..tons..

Correction was done.

line 250 Figure 1 – no bars are visible in Figure 1

Figure 1 was done again, and the bars were included.

Figure 1b – change the scale on the Y-axis from 0 to 40; the lines will be more visible.

The scale Y axis of Figure 1b was design from 0 to 50, since one of the values reached this high value.

lines 236, 237 is:.. 13.0..., should be:..13..

Correction was done.

line 257 is:.. to 20 N... – are you sure? See Figure 2b after 25 days T1, T2, T3 the values are higher than 20 N

To clear the idea, we have rewritten the sentence (lines 280-282):

'In this case, the corresponding final values ranged from approximately 15 N to 28 N for red, yellow, and green bell peppers, while the orange bell pepper had a firmness up to 35 N.'

line 295 is:.. were very similar. Data shown an average range from 0.06 to 0.15%...., should be: ...were very similar and ranged from 0.06 to 0.15%.

The manuscript was modified according to the reviewer's comments as follows (lines 319-321):

'With respect to the TA, there were no statistical differences among treatments, storage days, and temperatures (data not shown); however, it occurred in the variable color. In this case, the green bell peppers had the lowest TA values (from 0.03% to 1 %)'.

line 367 is:.. 60 mg/L..., should be:. 60 mg L-1

Correction was done.

Reviewer 2 Report

Abstract

_The authors should rewrite the abstract; it's pretty confusing. They should also check the content of lines 20-23 regarding colour; it has contradictory information. Attention.

materials and methods

_In the Materials and Methods chapter, the authors must present, in the place of the text they understand best, a table that shows the parameters determined in each of the different types of bell peppers.

_Line 88 - In sub-chapter 2.1, briefly describe the extrusion method applied.

_Line 112 - Add the formula used to determine the value of ΔE.

Results and discussion

__Line 218 - In point 3.1.1. the discussion of the results obtained in this work is missing.

_Line 293 - In point 3.1.5. A table with TA results is missing.

_Line 476 - Check table 3 because it is missing to identify the values "21" and "25" as "days".

Conclusion

_What was the conclusion of this work? Does it translate only into your environmental impact? The authors should be more precise in the conclusion.

Moderate editing of the English language

Author Response

Reviewer 2

Abstract

_The authors should rewrite the abstract; it's pretty confusing. They should also check the content of lines 20-23 regarding colour; it has contradictory information. Attention.

In according to the reviewer’s comments, the abstract section was modified as follows:

Compared to commercial polyethylene nets, the bell peppers kept in the biodegradable nets did not show notable differences with respect to color, weight loss, total soluble solids, and titratable acidity. However, there were significant differences (p < 0.05) in terms of phenols content, carotenoids (orange bell peppers), anthocyanins, and vitamin C, with an overall tendency to show higher content in those kept in PLA 60% / PBTA 40% / cactus stem flour 3% than commercial packaging.

Materials and methods

_In the Materials and Methods chapter, the authors must present, in the place of the text they understand best, a table that shows the parameters determined in each of the different types of bell peppers.

TA table was added according to the reviewer’s suggestion:

The biochemical assays carried out according to the bell pepper color are shown in Table 1.

_Line 88 - In sub-chapter 2.1, briefly describe the extrusion method applied.

Correction was done.

_Line 112 - Add the formula used to determine the value of ΔE.

Correction was done.

Results and discussion

__Line 218 - In point 3.1.1. the discussion of the results obtained in this work is missing.

The discussion section was rewritten as follows:

However, differences in ΔE were observed by the effect of the color of the fruit, the orange and yellow bell peppers being the ones that reached the highest values (ΔE=50). These differences could be due to the diversity of pigments that prevail in each of them, such as carotenoids and capsanthin in red bell peppers, and chlorophylls in green bell peppers.

_Line 293 - In point 3.1.5. A table with TA results is missing.

TA data were not included because there were no significant changes over time, regardless of the color of the pepper. In addition, the manuscript contains many Tables and Figures, so we omitted the Figure, highlighting in the text that data was not shown.

_Line 476 - Check table 3 because it is missing to identify the values "21" and "25" as "days".

Correction was done.

Conclusion

_What was the conclusion of this work? Does it translate only into your environmental impact? The authors should be more precise in the conclusion.

The conclusion stated what net would be more adequate to follow studying. There was nothing more to say.

The use of PLA 60% / PBAT 40% / cactus stem flour 3%-based nets during the storage of bell peppers, regardless of color cultivar, generally maintained the same quality as those kept in commercial nets. Occasionally, the nets increased the content of the bioactive compounds and reduced the presence of bacteria, fungi, and yeast. As expected, the storage temperature influenced the ripening of the bell peppers, accelerating this process during the four-day storage period at 30 ºC. As a postharvest packaging for bell peppers, the biodegradable packaging PLA 60% / PBAT 40% / cactus nopal flour 3%-based could be considered a viable option with regard to reducing the environmental impact and postharvest loss due to the excessive use of packaging agricultural plastics, in this case, for bell peppers.

Reviewer 3 Report

The work entitle THE EFFECT OF NETTING BAGS ON THE POSTHARVEST QUALITY, BIOACTIVE AND NUTRITIONAL COMPOUNDS, AND THE SPOILAGE MICROORGANISMS CONTENT OF BELL PEPPERS is interesting as an alternative to the packaging nets commonly used for bell peppers act as a form for protection. The biodegradable nets presented could be considered as a viable option for postharvest packaging for bell peppers.

The work affects the maintenance or improvement of commercial aspects such as color, weight loss, total soluble solids, titrable acidity, respiration rate and variables such as phenols, carotenoids, anthocyanins, chlorophyll, and vitamin C. Compare related aspects with the color of the peppers, the type of nets, and storage days which is very valid for postharvest packaging for bell peppers.

Despite the large amount of work, some errors have been detected that should be corrected:

-The titles of the results and discussion sections TSS and TA should be extended to make them clearer.

-Table 1 must adjust lines and paragraphs.

-Figure 3 is not clear, since the different plots correspond to the different net but the colors are different types of pepper (colors) or only one type has been analyzed?

-The figures that are presented in panels individualized according to the color of the pepper, would be more visible and understandable if each panel were the color of the pepper, for example in degraded tones or different patterns with the same color. In those that are done with only green or red, the same thing and thus it would be clearer when it is done with different ones and when with only one group.

- Figure 6 legend should be green bell peppers?

- The results obtained in green peppers in the count of bacteria and fungi in treatment 4 for 30ºC and 8ºC are surprising. Any explanation for this?

- Microbiological analysis results are not sufficiently described.

Author Response

Reviewer 3

The work entitle THE EFFECT OF NETTING BAGS ON THE POSTHARVEST QUALITY, BIOACTIVE AND NUTRITIONAL COMPOUNDS, AND THE SPOILAGE MICROORGANISMS CONTENT OF BELL PEPPERS is interesting as an alternative to the packaging nets commonly used for bell peppers act as a form for protection. The biodegradable nets presented could be considered as a viable option for postharvest packaging for bell peppers.

The work affects the maintenance or improvement of commercial aspects such as color, weight loss, total soluble solids, titrable acidity, respiration rate and variables such as phenols, carotenoids, anthocyanins, chlorophyll, and vitamin C. Compare related aspects with the color of the peppers, the type of nets, and storage days which is very valid for postharvest packaging for bell peppers.

Despite the large amount of work, some errors have been detected that should be corrected:

-The titles of the results and discussion sections TSS and TA should be extended to make them clearer.

Correction was done.

-Table 1 must adjust lines and paragraphs.

Correction was done.

-Figure 3 is not clear, since the different plots correspond to the different net, but the colors are different types of pepper (colors) or only one type has been analyzed?

Thank you for your thoughtful suggestion. The authors in the materials and methods section included the following information (lines 137-139):

'Because the nets were not tightly closed, experimental units of three fruit (red, orange, yellow, or green) were each placed in a sealed jar (vol. 4.0 L) and kept at room temperature (25 ± 2 °C) for 2 h. Analysis of this variable was not carried out in terms of color.'

-The figures that are presented in panels individualized according to the color of the pepper, would be more visible and understandable if each panel were the color of the pepper, for example in degraded tones or different patterns with the same color. In those that are done with only green or red, the same thing and thus it would be clearer when it is done with different ones and when with only one group.

Thank you for your suggestion; however, there was a problem with the software used, and it does not allow us to modify the Figures. We are sorry for that, and in future studies, we would like to follow your good advice to take them.

- Figure 6 legend should be green bell peppers?

Correction was done.

- The results obtained in green peppers in the count of bacteria and fungi in treatment 4 for 30ºC and 8ºC are surprising. Any explanation for this?

As suggested, we added more information:

'…this was probably due to an accelerated ripening process, their antioxidant and nutritional composition, and a major susceptibility compared to other colors.'

- Microbiological analysis results are not sufficiently described.

Results were rewritten as follows (lines 485-499):

'The presence of bacteria, fungi, and yeast was statistically different (p < 0.01) among color groups, treatments, and storage periods (Table 4). With respect to the bacterial count, regardless of stored temperature, there was a tendency in the bell peppers red, orange, and yellow to have the lowest microorganism count (from 3 to 20 times less) when stored in the nets-based PLA 60% / PBAT 40% / cactus stem flour 3% (T3) compared to bell pepper stored with PLA 60% / PBAT 40% (T2) and without nets (T1). Except for the green one, a low count was also observed in the bell peppers stored in T4. With regard to fungi and yeast, the lowest average counts corresponded to those fruit stored at 8 ºC in T3, with corresponding average values of 27 CFU. In the bell peppers stored in commercial nets, the final values were 53 CFU at 8 ºC and 43 CFU at 30 ºC. The orange and green bell peppers stored without nets had the highest fungi and yeast count (350 CFU and 299 CFU at 8 ºC, respectively). Finally, the green pepper packed in commercial nets (T4) showed a considerably higher difference in the bacteria and fungi counts compared to the remaining treatments. This was probably due to an accelerated ripening process, their antioxidant and nutritional composition, and a major susceptibility compared to other colors.'